# Oncologic Applications of Artificial Intelligence and Deep Learning Methods in CT Spine Imaging—A Systematic Review

**DOI:** 10.3390/cancers16172988

**Published:** 2024-08-28

**Authors:** Wilson Ong, Aric Lee, Wei Chuan Tan, Kuan Ting Dominic Fong, Daoyong David Lai, Yi Liang Tan, Xi Zhen Low, Shuliang Ge, Andrew Makmur, Shao Jin Ong, Yong Han Ting, Jiong Hao Tan, Naresh Kumar, James Thomas Patrick Decourcy Hallinan

**Affiliations:** 1Department of Diagnostic Imaging, National University Hospital, 5 Lower Kent Ridge Rd, Singapore 119074, Singapore; aric.lee@mohh.com.sg (A.L.); weichuan.tan@mohh.com.sg (W.C.T.); dominic.fong@mohh.com.sg (K.T.D.F.); david.lai@mohh.com.sg (D.D.L.); yiliang.tan@mohh.com.sg (Y.L.T.); xi_zhen_low@nuhs.edu.sg (X.Z.L.); shuliang_ge@nuhs.edu.sg (S.G.); andrew_makmur@nuhs.edu.sg (A.M.); shao_jin_ong@nuhs.edu.sg (S.J.O.); yonghan_ting@nuhs.edu.sg (Y.H.T.); james_hallinan@nuhs.edu.sg (J.T.P.D.H.); 2Department of Diagnostic Radiology, Yong Loo Lin School of Medicine, National University of Singapore, 10 Medical Drive, Singapore 117597, Singapore; 3National University Spine Institute, Department of Orthopaedic Surgery, National University Health System, 1E, Lower Kent Ridge Road, Singapore 119228, Singapore; jonathan_jh_tan@nuhs.edu.sg (J.H.T.); dosksn@nus.edu.sg (N.K.)

**Keywords:** artificial intelligence, deep learning, machine learning, spinal oncology, computed tomography imaging, applications

## Abstract

**Simple Summary:**

In recent years, advances in deep learning have transformed the analysis of medical imaging, especially in spine oncology. Computed Tomography (CT) imaging is crucial for diagnosing, planning treatment, and monitoring spinal tumors. This review aims to comprehensively explore the current uses of deep learning tools in CT-based spinal oncology. Additionally, potential clinical applications of AI designed to address common challenges in this field will also be addressed.

**Abstract:**

In spinal oncology, integrating deep learning with computed tomography (CT) imaging has shown promise in enhancing diagnostic accuracy, treatment planning, and patient outcomes. This systematic review synthesizes evidence on artificial intelligence (AI) applications in CT imaging for spinal tumors. A PRISMA-guided search identified 33 studies: 12 (36.4%) focused on detecting spinal malignancies, 11 (33.3%) on classification, 6 (18.2%) on prognostication, 3 (9.1%) on treatment planning, and 1 (3.0%) on both detection and classification. Of the classification studies, 7 (21.2%) used machine learning to distinguish between benign and malignant lesions, 3 (9.1%) evaluated tumor stage or grade, and 2 (6.1%) employed radiomics for biomarker classification. Prognostic studies included three (9.1%) that predicted complications such as pathological fractures and three (9.1%) that predicted treatment outcomes. AI’s potential for improving workflow efficiency, aiding decision-making, and reducing complications is discussed, along with its limitations in generalizability, interpretability, and clinical integration. Future directions for AI in spinal oncology are also explored. In conclusion, while AI technologies in CT imaging are promising, further research is necessary to validate their clinical effectiveness and optimize their integration into routine practice.

## 1. Introduction

Spinal malignancy encompasses both primary spine tumors and secondary spine metastases, with the latter being more prevalent [1,2]. The repercussions of spinal malignancy on quality of life are profound, stemming from complications like pain resulting from fractures (mechanical), spinal cord compression, and neurological deficits [3]. Moreover, it can lead to diminished mobility, bone marrow aplasia, and hypercalcemia, which manifest as symptoms such as constipation, polyuria, polydipsia, fatigue, and potentially life-threatening conditions like cardiac arrhythmias and acute renal failure [4,5]. Therefore, early detection, accurate diagnosis, and effective treatment of spinal malignancies, particularly metastases, are imperative to minimize complications and enhance patients’ overall well-being [6].

Artificial intelligence (AI) has made significant strides in the field of spinal oncology, offering radiologists valuable assistive tools to enhance their diagnostic capabilities and streamline workflows. In several applications, the findings in spinal oncology are either promising or have already surpassed previous benchmarks. For instance, the deep learning algorithm developed by Matohashi et al. [7] showed comparable sensitivity in detecting spinal metastases compared to orthopedic and radiology experts, and improved the sensitivity of in-training clinicians. AI has the potential to have a substantial impact on each step of the imaging value chain [8]. At this early stage in the integration of AI into clinical radiology, several studies have demonstrated its potential and utility. In recent years, there has been a growing interest in various AI applications, particularly in automating the detection and classification of spinal lesions [9]. CT, being a reliable imaging modality for assessing osseous involvement and the degree of destruction in spine abnormalities [10,11], has been widely studied for these reasons.

The purpose of this article is to introduce artificial intelligence techniques through a review of recent research, emphasizing their use at various phases of spinal oncology CT imaging, including image production and utilization. This narrative review aims to evaluate the current achievements of artificial intelligence (AI) and its possible applications to spinal oncology for both experts in the field and non-specialist readers. We hope to provide an overview of the most clinically pertinent applications of machine learning, including radiomics and convolutional neural networks (CNNs), in spinal oncology CT imaging, and to discuss potential future innovations in its use.

## 2. Technical Aspects

### 2.1. AI, ML, and DL

While “AI”, “machine learning” (ML), and “deep learning” (DL) are often used interchangeably, they actually represent distinct concepts. “AI” encompasses methods that enable computers to mimic intelligent human behavior [12,13]. “ML” is a specialized field within AI, utilizing statistical, mathematical, and computer science tools to enhance machine performance through experience [14,15,16]. “DL” further narrows down ML (Figure 1), focusing on the utilization of deep neural networks to analyze large datasets [17]. This particular domain is even more specialized compared to conventional machine learning, and leverages artificial neural networks, which involve several layers, to solve complex medical imaging challenges [18,19].

The complex layered structure allows the deep learning model or algorithm to learn information directly from imaging datasets and predict fundamental diagnostic patterns and features that surpass human capabilities [20,21]. These advanced deep learning techniques can then be used in accurate image classification (detecting the presence or absence of disease, as well as classifying disease severity), segmentation (based on individual pixels), and detection. Unlike traditional machine learning approaches that rely on manually engineered feature extraction from input images, deep learning methods autonomously learn image features by directly analyzing input images using multi-layer neural networks [22,23], which include convolutional neural networks (CNNs). This process transforms input images into valuable outputs. Deep learning systems not only excel at mapping image features to desired outputs but also possess the capability to discern intrinsic image details that often surpass human perception [24]. This advancement has significantly influenced the emergence of radiomics, a specialized field within medical imaging. Radiomics leverages various machine learning techniques, including deep learning, to extract and analyze a vast array of quantitative features from medical images. By utilizing these extracted features, radiomics enhances the identification, differentiation, and prognosis of lesions, enabling more precise differentiation between pathological and histological tumor subtypes. This capability underscores the pivotal role of deep learning in advancing radiomics, providing a deeper understanding of complex medical images and improving diagnostic accuracy [25].

### 2.2. Radiomics

Radiomics, an emerging field within AI and machine learning, involves converting radiological images containing important tumor-related information into quantifiable data [25]. These data assist clinicians in evaluating tumors beyond subjective visual interpretation, offering insights into tumor behavior and pathophysiology, including subtyping and grading [26,27]. When combined with clinical and qualitative imaging data, radiomics enhances medical decision-making processes, aiding disease prediction, prognosis assessment, and treatment response monitoring [28,29].

The workflow for building a radiomics model [30] comprises several steps: image acquisition, image processing (including segmentation and region of interest (ROI) selection), image feature extraction within the ROI, exploratory analysis and feature selection, and finally model building and classification (Figure 2). Factors such as the imaging modality, acquisition techniques, software, segmentation methods, and algorithm structure impact the process and vary accordingly [31,32]. Machine learning techniques like random decision forests can then be employed to validate and enhance the classification accuracy of predictors [33,34]. Ultimately, these tools can then be applied in the clinical setting to improve diagnostic accuracy and prognosis prediction [28,35,36,37,38].

Radiomic techniques consist of two main approaches: handcrafted-feature and deep learning-based analysis [39]. Handcrafted-feature radiomics [40,41] involves extracting numerical features from segmented regions of interest, categorized into shape, first-order statistics, textural features, and higher-order statistical features. Machine learning models, such as a Cox proportional hazards model, single vector machine (SVM), decision trees, and regression, are then developed to make clinical predictions based on these features [42,43] and validated for efficiency and sensitivity. In contrast, deep learning techniques use neural network architectures like CNNs to automatically extract important features from radiological images without prior descriptions [44,45,46]. These features undergo further processing for analysis or are used to generate machine learning models, which are then validated on larger datasets before they are applied in a clinical setting [47,48].

Although still in the early stage of development, the advent of radiomics has the potential to revolutionize medical imaging by offering rapid and comprehensive tumor characterization [49,50]. Radiomics holds the promise of enhancing traditional clinical testing and prognostication, making it a highly anticipated development for the future [51].

## 3. Materials and Methods

### 3.1. Literature Search Strategy

We systematically searched major electronic databases (PubMed, MEDLINE, Web of Science, clinicaltrials.gov) following PRISMA guidelines [52]. The search strategy involved using a combination of keywords and Medical Subject Headings (MeSH) to ensure comprehensive coverage. Specifically, we used the following query: (“Artificial intelligence” OR “AI” OR “deep learning” OR “machine learning” OR “convolutional neural network” OR “neural network” OR “radiomics”) AND (“spine” OR “spinal” OR “vertebral”) AND (“CT” OR “CT imaging”) AND (“malignancy” OR “metastasis” OR “cancer” OR “tumor” OR “oncology”). This query was applied to the title and abstract fields of the articles to ensure the retrieval of studies most relevant to our review topic. Two authors (W.O. and A.L.) independently reviewed the collected references and selected studies for detailed full-text screening. The literature search was conducted up to 31 March 2024.

### 3.2. Study Screening and Selection Criteria

In conducting this systematic review, we adopted an extensive search strategy to ensure a comprehensive evaluation of the literature. Our search process involved no specific restrictions, aiming to encompass a wide range of relevant studies. We focused on identifying scientific research that utilized radiomic techniques, artificial intelligence (AI), or deep learning methods within the context of CT imaging for spinal oncology.

To be included in the review, studies had to meet several criteria. First, the research needed to involve CT imaging analysis, specifically applying radiomics, AI, or deep learning techniques to the study of spinal oncology. Second, the studies had to assess the predictive and diagnostic capabilities of these technologies, with a particular emphasis on their integration into clinical practice and their potential to enhance diagnostic accuracy or provide prognostic insights. Third, only studies involving human subjects were considered to ensure the applicability of findings to real-world clinical settings. Finally, we limited our review to publications in English to maintain consistency and facilitate a thorough evaluation of the literature.

We excluded studies that did not align with these criteria. Specifically, case reports, editorial correspondence (such as letters, commentaries, and opinion pieces), and review articles were not included, as these do not present original research findings. Additionally, we excluded publications that focused on non-imaging radiomic techniques or did not utilize AI for the clinical analysis of CT images.

To ensure a thorough and exhaustive search, we concluded our literature review by examining the bibliographies of the selected publications. This additional step was undertaken to identify any relevant studies that might have been missed during the initial search, thereby enhancing the comprehensiveness of our review on the oncologic applications of AI and deep learning methods in CT spine imaging.

### 3.3. Data Extraction and Reporting

All chosen research articles were collected and organized into a spreadsheet using Microsoft Excel (Microsoft Corporation, Redmond, WA, USA). Information gathered from each research article included:Details of the research article: authors, publication date, journal name;Primary clinical application: detection, classification, segmentation, treatment, or prognosis prediction;Study specifics: study type, patient characteristics or imaging modalities, body parts scanned, and specific bone areas segmented for analysis (e.g., internal or external datasets);Machine learning methodologies utilized: radiomics, artificial neural networks, convolutional neural networks, etc.

## 4. Results

### 4.1. Search Results

The initial search across major electronic medical databases (Figure 3) identified a total of 226 relevant articles, which were screened based on the specified criteria. Articles were excluded if they were published more than 15 years ago, not written in English, were non-journal articles, or were conference abstracts/articles. Based on these exclusion criteria, 18 publications were initially excluded, leaving 208 articles for further full-text analysis to determine inclusion. For the remaining 208 articles, full-text reviews were performed, and a total of 177 publications were excluded as they focused on cancer sites other than the spine or did not utilize imaging data. An additional two articles were included after manually reviewing the bibliographies of selected articles. This process resulted in a total of 33 articles (Figure 1) selected for thorough analysis. Key findings from these studies were compiled (Table 1) and summarized in this review, with the main tasks of the AI models categorized into four broad categories: (1) Detection, (2) Classification, (3) Prognosis, and (4) Treatment Planning [53]. Most studies did not provide sufficient data to construct 2 × 2 contingency tables, precluding a formal meta-analysis.

Our search revealed that 12/33 studies (36.4%) focused on using AI techniques for detecting spinal malignancy, while 11/33 studies (33.3%) concentrated on classification. Additionally, 6/33 studies (18.2%) utilized radiomics for prognostication, 3/33 studies (9.1%) focused on treatment planning, and 1/33 study (3.0%) addressed both detection and classification. Among the studies focusing on classifying spinal malignancy using AI, 7/33 studies (21.2%) employed machine learning methods to distinguish between benign and malignant lesions, while 3/33 studies (9.1%) evaluated the stage or grade of spinal malignancy, and 2/33 studies (6.1%) classified the presence of certain tumor biomarkers using radiomic applications. Of the papers focusing on predicting prognosis, 3/33 studies (9.1%) dealt with predicting complications such as adverse outcomes or pathological fractures, while 3/33 studies (9.1%) aimed to predict treatment outcomes.

### 4.2. Performance Assessment

It is important to be aware of the various methodologies used to assess the performance of AI systems [54]. The most common key metrics used to assess the performance of AI systems include sensitivity, specificity, accuracy, precision, Area Under the Curve (AUC), Figure of Merit (FOM), F1-score [55,56], and kappa. Each of these metrics provides different insights into the performance of AI algorithms and their clinical relevance, with values closer to 1.0 indicating good performance:Sensitivity (Recall) and Specificity: sensitivity (recall) measures the proportion of true positives (spinal tumors correctly identified by the AI system) out of all actual positives (all spinal tumors present in the CT images) [57]. Specificity, on the other hand, quantifies the ability of the AI system to correctly identify true negatives (normal spinal conditions) out of all actual negatives (all non-pathological conditions). These metrics are essential to assessing how well AI algorithms detect both positive and negative cases in spinal oncology;Accuracy and Precision: accuracy indicates the overall correctness of the AI system’s predictions, measuring the ratio of correctly predicted cases (both true positives and true negatives) to the total number of cases evaluated. Precision, meanwhile, focuses on the AI system’s ability to accurately identify positive cases among all predicted positive instances, minimizing false positives. These metrics provide a comprehensive view of the AI algorithm’s reliability and correctness in clinical diagnosis;Area Under the Curve (AUC): AUC evaluates the performance of AI models in binary classification tasks [58,59], such as distinguishing between diseased and healthy spinal conditions based on CT imaging features. A high AUC value indicates that the AI model effectively ranks diseased cases higher than healthy ones, demonstrating its discriminatory power in spinal oncology diagnostics;Figure of Merit (FOM) and F1-score [60]: Figure of Merit encompasses a range of metrics including sensitivity, specificity, accuracy, and precision, tailored to the specific diagnostic challenges presented by spinal tumors. F1-score, a harmonic mean of precision and recall, balances the trade-off between these metrics and is particularly useful in scenarios where there is an imbalance between positive and negative cases in the dataset [61]. A model with a high F1 score indicates both good precision and recall, reflecting a robust model;The kappa statistic [62], often denoted as κ, is a measure used to assess the level of agreement between two or more raters or classifiers beyond what would be expected by chance alone. It is particularly useful in the context of classification tasks, where it evaluates how well the AI system’s predictions align with the true classifications compared to random chance [63]. Kappa values range from −1 to 1. A kappa value of 1 indicates perfect agreement between the raters, 0 indicates agreement no better than chance, and negative values suggest worse-than-chance agreement. High kappa values indicate that the AI system’s predictions are consistently aligned with expert assessments or ground truth, which is vital for the system’s clinical applicability and trustworthiness.

These metrics provide a framework for assessing the clinical relevance and performance of AI algorithms in interpreting CT scans. By quantifying these metrics, clinicians and researchers can effectively evaluate the diagnostic accuracy, reliability, and potential impact of these AI-driven approaches in enhancing spinal tumor detection and management.

**Table 1 cancers-16-02988-t001:** Key characteristics of the selected articles.

Authors	AI Method	Publication Year	Main Objectives	Journal	Main Task	Sample Size (No. of CTs/Patients)	Performance of AI Model
Tatjana W. et al. [64]	SVM classifier	2012	Using supervised learning methods to detect sclerotic bone metastases in CT imaging.	SPIE Medical Imaging	Detection	22	Sensitivity: 71.2–87.0%
Hammon M. et al. [65]	CADe (RF classifier)	2013	Automated detection of osteolytic and osteoblastic spine metastases on CT.	European Radiology	Detection	134	Sensitivity 83.0% (lytic), 88.0% (blastic)
Burns J. et al. [66]	CADe (SVM classifier)	2013	Automated detection of sclerotic metastases in the thoracolumbar spine on CT	Radiology	Detection	59	Sensitivity 79.0–90.0%
Roth. H. et al. [67]	CNN (DropConnect)	2014	Using Deep CNN methods to detect sclerotic spinal metastases on CT imaging.	Recent Advances in Computational Methods and Clinical Applications for Spine Imaging	Detection	59	Sensitivity: 79.0%; AUC 0.834
Masoudi S et al. [68]	ResNet-50 with DC-GAN augmentation	2020	Detect and classify bone lesions in CT images of prostate cancer patients.	Journal of Clinical Oncology	Detection and Classification (Benign vs. malignant)	56	Sensitivity: 81.0%; accuracy 89.0%
Fan X et al. [69]	AlexNet	2021	Using Deep Learning Methods to Identify Spinal Metas in Lung Cancer using CT Images.	Hindawi	Detection	36	Sensitivity: 66.0–81.4%
Noguchi S. et al. [70]	2D U-Net, 3D U-Net, and ResNet	2022	Deep learning to improve radiologists’ performance in spinal metastases detection on CT	European Radiology	Detection	732	Sensitivity: 82.4–89.6%; improved radiologist’s sensitivity by 15.3%; reduced mean interpretation time by 83 s (*p* < 0.05).
Hoshiai S. et al. [71]	DL Model	2022	Detecting vertebral bone metastases on CT	European Journal of Radiology	Detection	130	FOM from 0.848 to 0.876 (radiologist) and from 0.752 to 0.799 (resident), (*p* < 0.05 for both)
Musa A. et al. [72]	PyRadiomics	2022	Detecting prostate cancer bone/spinal metastases invisible in CT	Current Medical Imaging	Detection	53	Accuracy 85.0%; sensitivity 78.0–91.0%; specificity: 88.0–93.0%
Gilberg L. et al. [73]	DLA (U-Net)	2023	Deep learning to improve Radiologist’s detection of spinal malignancies in CT imaging.	Applied Sciences	Detection	32	Sensitivity: 75.0%; improved radiologist’s sensitivity by 20.8%
Huo T. et al. [74]	3D U-Net (DCNN)	2023	Deep learning to improve radiologists’ performance in lung cancer spinal metastases detection on CT	Frontiers in Oncology	Detection	126	Sensitivity: 89.4%; improved radiologist’s sensitivity by 22.2% and accuracy of 26.2%
Koike Y. et al. [75]	YOLOv5m and InceptionV3	2023	AI-aided lytic spinal bone metastasis detection on CT scans	International Journal of Computer Assisted Radiology and Surgery	Detection	2125	Accuracy 87.2%; precision 94.8%; recall: 74.1%; F1-score 74.1%; AUC 0.940
Motohasi M. et al. [7]	U-Net (DeepLabv3+)	2024	Using Deep Learning Algorithm to Detect Spinal Metastases on CT Images.	Spine	Detection	435	Sensitivity: 75.0–78.0%; precision: 36.0–68.0%; F1 score: 0.48–0.72
Chmelik J. et al. [76]	CNN	2018	Segmentation and classification of metastatic spinal lesions in 3D CT data	Medical Image Analysis	Classification (Benign vs. malignant)	31	AUC 0.780–0.800; sensitivity: 71.0–74.0%; specificity: 82.0–88.0%
Li Y. et al. [77]	ResNet50	2021	Differentiate benign and malignant vertebral fracture on CT using deep learning	European Radiology	Classification (Benign vs. malignant)	433	Sensitivity: 95.0%; specificity: 80.0%; accuracy: 88.0%
Masoudi S. et al. [78]	2D ResNet-50, ResNeXt-50, 3D ResNet-18, 3D ResNet-50	2021	Differentiate benign versus malignant spinal lesions on CT.	IEEE Access	Classification (Benign vs. malignant)	114	Accuracy: 79.4–92.2%; F1-Score: 0.755–0.923
Hallinan J. et al. [79]	R-CNN (ResNet50)	2022	Deep learning algorithm for grading cord compression secondary to spinal metastasis/epidural disease on CT	Cancers	Classification (Stage/Grade)	444	kappas (κ: 0.873–0.911); AUC: 0.953–0.971; sensitivity: 92.6–98.0%; specificity: 94.8–99.8%
Naseri H. et al. [80]	PyRadiomics	2022	Radiomics-based machine learning models to distinguish between metastatic and healthy bone.	Scientific Report	Classification (Benign vs. malignant)	170	AUC 0.640–0.950
Park, T. et al. [81]	U-Net (CNN)	2022	Automated segmentation of the fractured vertebrae on CT and using radiomics to predict benign versus malignant.	Nature	Classification (Benign vs. malignant)	158	Dice similarity coefficient: 0.930–0.940; AUC 0.800–0.930
Wang, Q. et al. [82]	Research Portal V1.1	2022	Using Machine learning techniques to predict RANKL expression of Spinal GCTB.	Cancers	Classification (Predict biomarkers)	107	AUC: 0.658–0.880 sensitivity: 65.7–97.9%; specificity: 23.3–71.9%; accuracy: 64.6–80.2%
Wang, Q. et al. [83]	PyRadiomics	2022	Clinical and CT-Based Radiomics based techniques to predict p53 and VEGF expression in Spinal GCBT.	Frontiers in Oncology	Classification (Predict biomarkers)	80	AUC 0.790–0.880
Hallinan J. et al. [84]	R-CNN (ResNet50)	2023	Deep learning method to diagnose epidural spinal cord compression using thoracolumbar CT.	European Spine Journal	Classification (Stage/Grade)	223	kappa (κ = 0.879); sensitivity: 91.8%; specificity: 92.0%; AUC: 0.919
Hallinan J. et al. [85]	R-CNN (ResNeXt50)	2023	Assess for metastatic spinal cord compression (mainly epidural extension) on CT imaging with external validation	Frontiers in Oncology	Classification (Stage/Grade)	420	Almost-perfect inter-rater agreement (κ = 0.813); sensitivity: 94.0%
Duan S. et al. [86]	Inception_V3	2023	Differentiating benign and malignant vertebral compression fracture on spinal CT imaging.	European Journal of Radiology	Classification (Benign vs. malignant)	280	AUC: 0.890–0.990; accuracy: 88.0–99.0%
Gui C. et al. [87]	PyRadiomics	2021	Radiomic modeling to predict risk of vertebral compression fracture after SBRT for spinal metastases	Journal of Neurosurgery	Prognosis (Predicting complications)	74	Sensitivity: 84.4%; specificity 80.0%, AUC 0.844–0.878 of 0.844, specificity of 0.800
Wang, Q. et al. [88]	PyRadiomics	2021	Using Radiomics-based technique to predict recurrence in spinal GCBT from pre-operative CT imaging.	Journal of Bone Oncology	Prognosis (Predict treatment outcome)	62	Accuracy: 89.0%; AUC 0.780 (predicting recurrence)
Massaad E. et al. [89]	CNN (Densenet and U-Net)	2022	Using machine learning methods to derive body composition analysis to predict complications in spine tumor surgery.	Journal of Neurosurgery Spine	Prognosis (Predict treatment outcome)	484	Body composition analysis using machine learning help predict risk for inferior outcomes.
Seou Y. et al. [90]	PyRadiomics	2023	Radiomics-based prediction of vertebral compression fracture prior to spinal SBRT from planning CT.	European Spine Journal	Prognosis (Predicting complications)	85	Accuracy: 78.8–82.9%; precision: 60.0–62.5%; F1 score: 0.573–0.650
Delrieu L. et al. [91]	U-Net	2024	Automated body composition analysis using L3 as reference on CT scans to predict treatment outcome in cancer patients.	Frontiers in Nuclear Medicine	Prognosis (Treatment Outcome)	352	DICE similarity coefficient: 0.850–0.940
Khalid S. et al. [92]	DL Model	2024	Detection of sarcopenic obesity and association with adverse outcomes in patients undergoing surgical treatment for spinal metastases	Journal of Neurosurgery Spine	Prognosis (Predicting complications)	62	DL detected sarcopenia patients increased odds of non-home discharge, readmission, and postoperative mortality.
Sebastiaan et al. [93]	DeepMedic	2022	Clinical Utility of CNN for treatment planning in spinal metastases.	Physics and Imaging in Radiation Oncology	Treatment Planning	782	DSC 96.7% HD: 3.6 mm. Acceptable: 77.0%
Netherton T. et al. [94]	CNN (X-Net)	2022	Automating Treatment Planning for Spinal Radiation Therapy using CT Imaging	International Journal of Oncology, Biology and Physics	Treatment Planning	220	Dice-similarity coefficient: 0.850 (cervical), 0.903 (thoracic), 93.7 (lumbar); AUC: 0.820; end-to-end treatment planning time <8 min
Hernandez S. et al. [95]	nn-UNet	2023	Automating Treatment Planning for Paediatric Craniospinal Radiation Therapy using CT.	Paediatric Blood Cancer	Treatment Planning	143	Dice similarity coefficient: 0.650–0.980; end-to-end treatment planning time: 3.5 ± 0.4 min

Single Vector Machine (SVM), Convolutional Neural Network (CNN), Deep Learning Algorithm (DLA), Random Forest (RF), Computer-Aided Detection (CADe), Deep Learning (DL), Dice Similarity Coefficient (DSC), Hausdorff Distance (HD), Figure of Merit (FOM), Area Under Curve (AUC), Receiver Operating Characteristic (ROC), Giant Cell Bone Tumour (GCBT).

### 4.3. Applications

#### 4.3.1. Detection of Spinal Lesions

Early identification and diagnosis of spinal lesions is crucial in clinical practice [96,97]. This plays a pivotal role in determining the disease stage in patients with malignancy, which significantly influences the treatment strategies and prognosis [98]. Spinal lesions, including metastases, are associated with increased morbidity, with over half of patients needing radiotherapy or invasive procedures due to complications such as spinal cord or nerve root compression [99,100,101]. Therefore, early diagnosis and targeted treatment before permanent neurological and functional deficits develop, is critical to achieve favorable outcomes [102,103,104].

However, detecting spinal lesions manually via various imaging techniques is often time-consuming, laborious, and challenging [105] due to factors such as overlapping imaging features with other pathologies [106] and variations in imaging quality, resulting in artifacts which can obscure lesions and hinder their detection [10,107]. Automated lesion detection is widely recognized for its potential to enhance radiologists’ sensitivity in identifying osseous metastases. Computer-aided detection (CADe) software and AI models have been shown to be as effective, if not more effective, than manual detection by radiologists.

The first CADe study using CT for spinal metastasis detection was conducted by O’Connor et al. in 2007 [108], specifically targeting osteolytic spinal metastases. This pioneering work laid the foundation for subsequent CADe research, which has been expanded to include other types of spinal metastases, such as osteoblastic and mixed lesions. Recently, advancements in artificial intelligence, particularly in DL and CNNs, have significantly enhanced the accuracy of CADe in detecting spinal metastases. These improvements have led to notable reductions in both false positive and false negative rates, thereby increasing the reliability of automated detection across different imaging techniques [67,69,109,110].

Several efforts employing diverse methods have been undertaken to detect spinal lesions in CT scans using AI techniques. For instance, Burns et al. [66] used a combination of watershed segmentation and a support vector machine classifier, achieving sensitivity of 79.0–90.0% in detecting osteoblastic spinal metastases. Hammon et al. [65] employed sequences of random forest classifiers that analyze local image features to evaluate regions for spinal metastases, attaining sensitivity of 83.0% for detecting osteolytic lesions and 88.0% in detecting osteoblastic lesions. Matohashi et al. [7] utilized deep learning methods using the “Deep Lab V3+” segmentation model, attaining sensitivity of 75.0–78.0% in detecting osteolytic lesions in thoracolumbar spine CT, which is comparable to that of orthopedic or radiology experts, and superior to that of in-training clinicians.

Several other techniques have been utilized to further improve sensitivity for detecting spinal metastases from CT. For instance, Hoshiai et al. [71] investigated the application of temporal subtraction CT for bone segmentation, using a multi-atlas-based method combined with spatial registration of two images via advanced computational techniques and deep learning methods showing improved overall field of merit (FOM) compared to both a board-certified radiologist and resident groups (*p* < 0.05). Both Noguchi et al. [9] and Huo et al. [59] developed deep convolutional neural network (DCNN) models that showed statistically significant improvements (*p* < 0.05) in the figure of merit (FOM) from 0.746 to 0.899 for radiologists, and in the detection accuracy and sensitivity of in-training radiologists, while also reducing the average interpretation time per case.

AI has already demonstrated its potential in oncology for improving the detection of various cancers. For example, in breast cancer screening, AI systems can accurately identify suspicious lesions on mammograms [111,112,113], reducing the workload for radiologists while maintaining high sensitivity and lowering recall rates [114,115]. In lung cancer, AI algorithms analyze CT scans to detect early-stage nodules with precision, facilitating timely intervention [116,117,118]. Additionally, AI tools in prostate cancer aid the detection of malignant tissues on MRI, improving diagnostic accuracy [119] and aiding treatment planning [119,120,121]. Although AI applications for detecting spinal lesions on CT scans are still in their infancy, their potential integration into clinical practice holds promise for helping alleviate radiologists’ workload [122,123] and reducing error rates [16]. By serving as a secondary reviewer, AI can enhance diagnostic accuracy and efficiency, providing a dependable safety net. Additionally, with improved detection accuracy and sensitivity, AI has the potential to minimize the necessity for further imaging modalities such as MRI or bone scans [124,125], potentially curbing healthcare costs.

#### 4.3.2. Classification of Spinal Lesions

AI has shown promise in differentiating benign versus malignant lesions across various cancers, leveraging advanced algorithms to improve diagnostic accuracy [126,127,128,129]. In the context of spinal lesions, machine learning has been applied to differentiate between various spinal pathologies, such as metastases, by identifying key radiomic features in vertebral lesions and incorporating these into diverse machine learning models [130]. Spinal lesion classification has been predominantly studied on MRI due to its superior soft tissue contrast and various sequences allowing for more precise characterization of lesion morphology and signal characteristics [131,132,133,134]. For example, Liu et al. [135] and Xiong et al. [136] utilized MRI-based radiomics to differentiate between multiple myeloma and spinal metastases. Yin et al. [137] came out with a radiomics model for the differentiation of primary Chordoma, Giant Cell Tumor, and metastatic sacral tumors.

Distinguishing benign from malignant spinal lesions on CT remains challenging. The complexity of spinal anatomy, the variability of lesion appearance, and the overlapping features between benign and malignant lesions pose significant hurdles [138,139]. AI models can potentially overcome these challenges to provide reliable differentiation through extensive training on diverse datasets and continuous refinement to improve accuracy and reduce false positives. For example, the deep learning method developed by Masoudi et al. [78], using lesion-based average 2D ResNet-50 and 3D ResNet-18 architectures with texture, volumetric, and morphologic information, achieved an accuracy of 92.2% for the classification of benign versus malignant sclerotic bony lesions in patients with prostate cancer. Chmelik et al. [76] developed a deep CNN-based segmentation and classification of difficult-to-define spinal lesions using 3D CT data, achieving an AUC of 0.780–0.800 in distinguishing metastatic (both osteoblastic and osteolytic) bone lesions versus benign bone lesions.

Differentiating vertebral fractures due to benign causes, such as osteoporosis, from those caused by malignancy on CT imaging poses significant challenges. Both types of fracture can present with similar radiologic features, including vertebral body collapse and cortical disruption, complicating accurate diagnosis [140]. Often, CT imaging alone fails to reliably distinguish these conditions, necessitating further work-up such as contrast-enhanced MRI or isotope bone scans [141,142]. AI, leveraging radiomics and deep learning techniques, offers a promising solution by analyzing complex patterns in CT images that are not easily discernible to the human eye [143]. Radiomics and deep learning models can be trained to identify subtle differences in image characteristics, such as bone density and lesion shape, and by integrating these advanced analytics AI can improve the differentiation between benign osteoporotic fractures and pathological vertebral fractures.

For example, Li Y. et al. [77] developed a deep learning model using a ResNet 50 network with ten-fold cross-validation to achieve 85.0–88.0% accuracy in distinguishing benign from malignant compression fractures. Duan S. et al. [86] came up with radiomics and deep learning methods using Inception_V3 to differentiate these two entities through CT features and clinical characteristics, achieving an AUC of up to 0.990. Both studies also concluded that visual features such as the presence of a soft tissue mass and bone destruction were highly suggestive of malignancy, while the presence of a transverse fracture line is highly suggestive of a benign fracture—consistent with the literature [140,144,145].

AI applications go beyond mere tumor detection and differentiation, and they can also automatically generate significant parameters, such as grading spinal lesions and their complications. For example, Hallinan et al. [79] employed a deep learning model to automate the classification of metastatic epidural disease and/or spinal cord compression on CT scans based on the Bilsky classification [101,103]. The model showed near perfect agreement when compared to trained radiologists, with kappas of 0.873–0.911 (*p* < 0.001). Subsequent studies by the same author [85] showed that their deep learning algorithm for metastatic spinal cord compression on CT showed superior performance to the CT report with almost-perfect inter-rater agreement (κ = 0.813) and high sensitivity (94.0%) as compared to CT reports issued by experienced radiologists which had only slight inter-rater agreement (κ = 0.027) and low sensitivity (44.0%) (*p* < 0.001). Precise and consistent classification of metastatic epidural spinal cord compression will help clinicians determine whether initial treatment should involve radiotherapy or surgical intervention [146]. These studies demonstrate the potential of deep learning to assist clinicians in the early diagnosis and grading of metastatic spinal disease, ensuring that appropriate treatment is delivered promptly.

Radiogenomics, which combines “Radiomics” and “Genomics”, entails utilizing imaging traits or substitutes to identify genomic patterns and advanced bio-markers within tumors [147,148]. These markers subsequently inform clinical decisions, encompassing the prognosis, diagnosis, and predictive accuracy of tumor subtypes [149]. The typical workflow of a radiogenomics study (Figure 4) encompasses five key stages: image acquisition and pre-processing, feature extraction and selection from both medical imaging and genotype data, association of radiomic and genomic features, data analysis utilizing machine learning models, and the establishment of a final radiogenomics outcome model [150].

In the realm of oncologic spinal CT imaging, the focus of radiogenomics research has predominantly been on giant cell tumor of the bone (GCTB). Wang et al. [82,83] pioneered the application of machine learning and radiomic techniques to forecast the presence of key biomarkers—RANKL (receptor activator of the nuclear factor kappa B ligand), p53, and VEGF (vascular endothelial growth factor)—in spinal GCTB, achieving an AUC of 0.658–0.880. Elevated levels of RANKL, p53, and VEGF expression [151] in GCTB have been correlated with more aggressive tumor behavior and increased recurrence risk [152,153,154]. These biomarkers also form the basis of targeted molecular therapies, such as Denosumab, a monoclonal antibody targeting RANKL, used to treat challenging-to-operate spinal GCTB cases [155,156]. Identification of these biomarkers aids in prognosis prediction and facilitates the selection of optimal disease management strategies. However, their assessment typically necessitates invasive tissue biopsies. The ability to predict these biomarkers non-invasively offers a quantitative and convenient approach to support predictive decision-making, ultimately enhancing patient outcomes [157].

#### 4.3.3. Prognosis and Predicting Complications

The integration of artificial intelligence (AI), deep learning, and radiomic models into CT spinal imaging holds promise for predicting outcomes and anticipating treatment complications in oncologic scenarios [158,159]. In addition to tumor characteristics, the assessment of sarcopenia and body composition plays a crucial role in the prognosis of patients with spinal metastases [160,161,162]. However, manual evaluation of these parameters can be tedious and exhausting, often requiring significant time and expertise [163]. AI and deep learning algorithms offer a transformative solution by automating the analysis of body composition from CT scans, providing rapid and accurate assessments [164]. These technologies can quantify muscle mass, adipose tissue distribution, and other relevant parameters with high precision, facilitating risk stratification and treatment planning.

For instance, Delrieu L et al. [91] developed a deep learning algorithm to automatically detect the L3 vertebra and segment body tissues to evaluate body composition and sarcopenia, achieving a median DICE similarity coefficient of up to 0.940 in relation to the sarcopenia metrics of the patient pool. Khalid et al. [92] came out with a machine learning-based technique for detection of sarcopenic obesity (SO) using CT in patients undergoing surgery for spinal metastases. Their algorithm effectively identified patients with SO, who exhibited increased odds of non-home discharge, re-admission, and post-operative mortality. These advancements enable clinicians to potentially recognize patients requiring nutritional interventions prior to invasive procedures, thereby enhancing overall treatment outcomes [165].

In addition to assessing body composition and sarcopenia, artificial intelligence (AI) holds promise for predicting the risk of recurrence, a crucial aspect in determining treatment outcomes for patients with spinal metastases. By analyzing radiomic features extracted from CT spinal imaging data, AI algorithms can identify subtle patterns and biomarkers associated with tumor aggressiveness and likelihood of recurrence [166,167]. For example, Wang Q et al. [88] developed a deep learning model to predict the risk of recurrence of spinal GCTB using images from pre-operative CT, achieving an accuracy of 89.0% and an AUC of 0.780. These predictive models leverage deep learning techniques to process vast amounts of imaging data and generate personalized risk assessments for individual patients [168]. Early identification of patients at high risk of recurrence enables clinicians to tailor treatment strategies, such as implementing adjuvant therapies or closer surveillance protocols, to mitigate the risk of disease progression and improve long-term outcomes [169].

Furthermore, AI-driven predictive models provide valuable insights into the underlying biological mechanisms driving tumor recurrence, facilitating the development of targeted therapeutic interventions aimed at preventing disease relapse and enhancing patient survival rates [170]. For instance, Gong et al. [171] came out with a radiomics model to predict pre-operative expression of PD-1 and PD-L1 in hepatocellular carcinoma (HCC), which was associated with more aggressive tumor behavior in the form of increased recurrence and distal metastatic risk [172]. The ability to use such an imaging biomarker may potentially help identify patients who will benefit from immune checkpoint inhibitor (ICI)-based treatment before surgery. Similarly, Bove et al. [173] developed a predictive model using radiomic features extracted from pre-treatment CT scans to forecast the likelihood of recurrence in patients with non-small cell lung cancer (NSCLC). Their findings demonstrated that specific radiomic signatures from the peri-tumor region correlated significantly with higher recurrence rates. By harnessing the power of AI-driven predictive models in spinal oncology, there is potential for clinicians to better stratify patients based on their risk of recurrence, tailor treatment strategies accordingly, and ultimately improve long-term survival rates.

AI and deep learning methodologies offer promising avenues to enhance the identification of patients at high risk of radiation-induced vertebral compression fracture (VCF) in the context of stereotactic body radiation therapy (SBRT) for spinal metastases. By extracting radiomic features from pre-treatment CT imaging data, these advanced techniques can provide more nuanced insights into the underlying tissue characteristics and microenvironment, enabling more accurate prediction of VCF risk [174]. Gui et al. [87] developed a machine learning model based on clinical characteristics and radiomic features from pre-treatment CT imaging to predict the risk of vertebral fracture following radiation therapy. Their model achieved sensitivity of 84.4%, specificity of 80.0%, and an area under the receiver operating characteristic (ROC) curve (AUC) of 0.878. The model developed by Seol et al., using similar methods, achieved an accuracy of 81.8% with an AUC of 0.870 in predicting VCF prior to spinal SBRT using pre-treatment planning CT. Integrating these AI-driven predictive models into clinical practice could potentially assist clinicians in implementing timely prophylactic measures to mitigate the occurrence of VCF and minimize associated morbidities before treatment [175]. This personalized approach to patient care will help optimize treatment strategies and improve overall treatment outcomes in individuals undergoing SBRT for spinal metastases.

#### 4.3.4. Treatment Planning

Accurate delineation of spinal lesions is paramount for the effective planning of radiotherapy for spinal neoplasms and metastases [176,177,178]. However, manual outlining often proves time-consuming and susceptible to variability between observers. Deep learning methods present a transformative approach by automating the identification and segmentation of spinal metastases and surrounding anatomical structures [179]. This automation not only facilitates precise radiation dose calculation and treatment plan optimization but also significantly streamlines the workflow [180,181]. By reducing manual intervention and mitigating inter-observer variability, deep learning expedites the treatment planning process, offering potential benefits in clinical settings [182,183]. 

For instance, Hernandez et al. [95] developed a comprehensive automated contour and planning tool for 3D-conformal craniospinal irradiation therapy using CT images for pediatric patients with medulloblastoma. Their model achieved remarkable DICE similarity coefficients (DSC) ranging from 0.650 to 0.980, with an average end-to-end treatment planning time of 3.5 ± 0.4 min. Similarly, Sebastiaan et al. [93] devised a CNN-based model for automating segmentation and delineation of vertebral bodies in radiotherapy treatment planning for spinal metastases. Their model exhibited an average computational time of less than 5 min compared to an average of 20 min for manual contouring, with DSC scores ranging from 0.944 to 0.967, and 77.0% of cases deemed clinically acceptable. Although still in its infancy and not yet fully validated in clinical settings, these studies underscore the potential of AI and deep learning methods for enhancing the efficiency of radiotherapy workflows without compromising accuracy, ultimately improving patient care.

In current clinical practice, deep learning tools such as RayStation by RaySearch Labs are actively used for tumor segmentation in gamma radiotherapy planning. For instance, Riguad et al. [184] utilized RayStation for optimizing dose planning in cervical cancer treatment, while Almberg et al. [185] applied it in breast cancer treatment, demonstrating significant improvements in dose optimization. These examples highlight the real-world effectiveness of advanced segmentation technologies. Additionally, other commercial solutions, such as Varian’s Eclipse [186] and Elekta’s Monaco [187], also employ sophisticated deep learning algorithms for radiotherapy planning.

## 5. Discussion

### 5.1. Interpretation and Implications of Findings

In our systematic review of oncologic applications of artificial intelligence (AI) and deep learning methods in CT spine imaging, we observed a broad spectrum of applications including detection, classification, prognosis, and treatment planning. Our review indicated that AI models show considerable promise in these areas, demonstrating relatively reliable performance and considerable improvement in clinical practice. However, several critical insights and interpretations emerged from our findings.

AI models have proven effective in enhancing the detection and classification of spinal lesions. Despite these advancements, the performance metrics reported across studies exhibit notable variability, reflecting a significant limitation. This variability is partly due to the fact that many studies were constrained by small sample sizes and were conducted at single centers. Such limitations raise concerns about the generalizability of the results [188]. Specifically, while AI models may perform well within specific study settings, their effectiveness and reliability across diverse clinical environments remain uncertain without broader validation [189]. In the realm of prognosis and treatment planning, AI models have demonstrated substantial potential by improving the prediction of patient outcomes and aiding in the formulation of treatment strategies. Nonetheless, translating these capabilities into routine clinical practice requires further validation. The current evidence base, constrained by the limited scope of existing studies, necessitates larger, multi-center trials to confirm the models’ effectiveness and ensure their applicability to a wider range of patient populations.

Overall, while our review underscores the potential of AI and deep learning in advancing CT spine imaging, addressing the identified limitations through comprehensive validation and integration efforts is crucial for realizing their full clinical potential. These steps will ensure that AI technologies can provide reliable, actionable insights, and support the management of spinal malignancies.

### 5.2. Integration into Clinical Practice

The integration of AI models into clinical workflows presents several significant challenges that must be carefully addressed. Current research often suffers from a lack of external validation and does not fully tackle the practicalities of implementing these models in routine clinical settings [190,191]. To address these barriers, future research should prioritize the standardization of imaging protocols to reduce variability [192], expand datasets through collaborative and multi-center studies, and develop AI systems that seamlessly integrate into existing clinical practices [193]. Additionally, obtaining FDA and CE approvals for these tools is a critical step, ensuring they meet rigorous safety and efficacy standards before widespread adoption. To date, only a limited number of AI applications in CT spinal oncology have received such approvals, with most focusing on treatment planning [185] in general, and not specific to spinal oncology. However, we anticipate that more AI tools will achieve regulatory clearance in the future as the technology matures and more evidence of its clinical utility emerges.

Furthermore, the integration of AI and machine learning (ML) into healthcare introduces a range of ethical concerns. These include issues related to privacy and data security, the risk of biases in AI algorithms, and the challenges of ensuring transparency and explainability of AI systems [194]. Additional concerns involve patient consent, autonomy, and the equitable access to these technologies [195]. The complexity of these issues is compounded by problems such as limited data availability, data drift, and the need for ongoing retraining and regulatory updates [196]. Ensuring the safety and clinical validation of AI tools is essential to prevent misdiagnoses, avoid inappropriate treatments, and mitigate potential health disparities. Therefore, a balanced approach is crucial to leverage the benefits of AI while addressing these significant ethical and practical challenges effectively.

### 5.3. Other Potential Applications

#### 5.3.1. Improving Image Quality

Oncologic spinal imaging poses significant challenges, particularly in complex cases such as post-surgical patients with metallic implants, which induce artifacts and degrade image quality, thereby impeding accurate evaluation of spinal lesions. These artifacts can obscure critical details (e.g., mass effect on the spinal cord or collections), complicating diagnosis, and treatment planning. However, deep learning reconstruction methods have emerged as valuable tools for enhancing CT imaging quality in oncology. Numerous studies have demonstrated their efficacy in mitigating artifacts and improving image clarity. For instance, Arabi et al. [197] developed a deep learning-based reconstruction algorithm that effectively reduced metal artifacts in PET/CT scans of post-surgical spinal patients, enhancing visualization of adjacent structures and facilitating more accurate lesion assessment. Similarly, Rui et al. [198] employed a deep learning-based metal artifact correction (MAC) algorithm, achieving significantly higher subjective scores as compared to conventional MAC and virtual monochromatic imaging (VMI) techniques.

Building upon these successes, there is potential for Generative Adversarial Networks (GANs) to further advance the field. GANs offer the ability to generate realistic and artifact-corrected images by learning from large datasets, potentially providing a solution to the challenges posed by various artifacts in CT spinal imaging. For instance, Lu et al. [199] utilized GAN to correct motion artefacts on CT Coronary Angiogram images. Their GAN-generated images showed statistically significant improvement in motion artifact alleviation score (4/5 vs. 1/5, *p* < 0.001) and overall image quality score (4/5 vs. 1/5, *p* < 0.001), with high accuracy in identifying stenosis (81.0% vs. 66.0%) in the mid-right coronary artery as compared to motion-affected images. Goli et al. [200] utilized GAN to improve CT images of the head and neck affected by metallic artefacts from dental implants, achieving 16.8% improvement in assessing the oral cavity region, which is important for treatment planning. While traditional non-AI-based metal artifact correction methods, such as i-MAR used in Siemens CT scanners [201,202], are commonly employed in clinical settings, deep learning reconstruction methods offer several significant advantages. These methods enhance image quality by more effectively reducing metal artifacts and improving the visualization of adjacent structures [203]. They provide greater adaptability to new data and imaging modalities, reduce the need for manual adjustments, and often result in faster processing times and better quantitative accuracy [204]. Consequently, deep learning techniques offer potentially more robust solutions for addressing metallic artifacts in CT imaging, leading to more reliable and accurate diagnostic outcomes [205]. By synthesizing artifact-free images, GANs hold promise for improving diagnostic accuracy and enhancing oncologic evaluations in patients with spinal lesions.

#### 5.3.2. Predicting Primary Malignancy from Spinal Metastases

Spinal metastasis originating from an unidentified primary tumor presents a common clinical challenge, affecting up to 30% of patients on initial presentation [206,207]. While conventional CT scans can accurately detect vertebral metastases, distinguishing between cancers of various origins can be challenging as they often appear similar. Consequently, these patients often have to undergo additional PET/CT imaging for primary cancer diagnosis and comprehensive whole-body staging scans prior to treatment initiation [208]. In rare cases, even with further imaging, the primary tumor remains unidentified, necessitating an invasive biopsy to determine the probable primary tumor site and to explore more targeted treatment options.

Several studies have demonstrated the effectiveness of deep learning models in predicting primary tumor sites in patients with spinal metastases using MRI-derived imaging features. For example, Liu et al. [209] explored the feasibility of a ResNet-50 convolutional neural network model for this purpose, achieving an AUC–ROC of 0.770 and 53.0% accuracy in classifying spinal metastases originating from the lung, kidney, prostate, breast, and thyroid. Lang et al. [210] utilized radiomics and deep learning techniques to distinguish spinal metastases from lung cancer and other origins using dynamic contrast-enhanced (DCE) sequences from a spinal MRI database. Their findings indicated that the DCE kinetic measurement of the washout slope from a hotspot within the spinal metastatic lesion was the most reliable parameter for diagnosing primary lung cancer versus other tumor types, achieving accuracies of up to 81.0%.

Cao et al. [211] investigated radiomics for distinguishing primary tumors from brain metastases using CT images. Shang et al. [184] developed a similar model for identifying primary tumor types from lung metastases in thoracic CT scans. Despite bone metastases being common, no studies have specifically addressed primary tumor differentiation in spinal metastases using CT imaging. This gap presents a significant opportunity for future research in spinal oncology. With promising results seen in MRI studies, applying deep learning techniques to CT spinal imaging shows potential. CT scans offer widespread availability and detailed anatomical information, making them valuable for assessing primary malignancies. By training deep learning models on large datasets of CT spinal images, robust algorithms could potentially differentiate and identify primary malignancies solely from CT scans. Leveraging radiomic approaches from other metastatic contexts could assist with developing predictive models specific to spinal metastases. Advanced imaging analytics applied to CT scans could reveal distinct patterns and biomarkers for different primary tumors, enhancing diagnostic accuracy and treatment planning. These advancements may provide clinicians with a non-invasive method to efficiently diagnose and plan treatments for spinal metastases.

#### 5.3.3. Quantifying Tumor Burden to Predict Treatment Response

Artificial intelligence (AI) applications hold promise for enhancing the measurement of tumor burden for assessing treatment response and monitoring tumor progression. AI-assisted segmentation allows for precise lesion and/or tumor volumetry, facilitating more accurate evaluations. For instance, Goehler et al. [212] implemented a deep learning method to estimate overall tumor burden for neuroendocrine neoplasia on MRI, achieving a concordance of 91% with manual clinician assessment and DSC of up to 0.81. Belal et al. [213] developed a fully automated CNN-based model to calculate skeletal tumor burden in patients with prostate cancer on PET/CT, achieving a moderately strong PET index correlation with that estimated by the physician (mean r = 0.69).

Assessing tumor burden and volume in spinal metastases presents challenges due to variations in vertebral shape and involvement across multiple levels [185] Manual volumetric assessment is not only time-consuming but also prone to variability among different observers, and even within the same observer. In clinical practice, evaluating radiological images by oncology specialists and radiologists is hindered by the labor-intensive nature of manual analysis, the absence of standardized quantification, and challenges related to reproducibility in both evaluation and measurement. Despite that, assessing tumor burden in spinal metastases is pivotal for predicting prognosis and treatment efficacy [214]. While there are studies which have looked into using deep learning to assess bone tumor burden in PET/CT [213,215] and bone scintigraphy [216], the current literature lacks studies exploring the use of CT imaging to comprehensively evaluate spinal tumor burden. This gap is significant, especially given the frequent use of whole-body CT scans in oncology for monitoring treatment response. Spinal metastases, including those affecting the spine, necessitate a holistic assessment of the entire skeleton due to the systemic nature of metastatic disease.

Recent advancements in deep learning applied to CT imaging have automated the segmentation of bone metastases, including those affecting the spine. Studies such as those by Motohashi et al. [7] and Saeed et al. [217] have shown deep learning algorithms’ effectiveness in identifying and quantifying metastatic lesions across skeletal regions. Despite these capabilities in segmenting bone tumors, there remains a lack of research specifically focused on assessing tumor burden. These AI-driven approaches could streamline the volumetric evaluation of spinal metastases and pave the way for improved clinical management by providing a comprehensive view of tumor burden throughout the body. Integrating AI into the assessment of bone metastases using CT holds the potential to significantly enhance the timely and accurate monitoring of treatment response. By automating tumor burden quantification from whole-body CT scans, AI can potentially guide oncologists to more effectively evaluate overall disease burden, which is crucial for assessing treatment efficacy and making informed decisions in patient care [218,219].

### 5.4. Study Limitations

Due to the heterogeneity among the studies reviewed and insufficient data, a formal meta-analysis could not be conducted. Consequently, our review is presented as a descriptive analysis rather than as a quantitative synthesis. This limitation affects our ability to perform statistical aggregation of results and derive more generalized conclusions. Additionally, the quality of data in the individual studies was not analyzed in detail, which could influence the robustness of the findings. However, this does not diminish the value of our review. Despite these limitations, our descriptive review provides valuable insights into the current state of oncologic applications of artificial intelligence and deep learning methods in CT spine imaging. It highlights key trends, identifies research gaps, and offers a comprehensive overview of technological advancements and their clinical implications. By synthesizing and summarizing findings from a diverse range of studies, our review contributes significantly to the understanding of these emerging technologies and provides a foundation for future research directions and clinical applications in the field. The qualitative synthesis presented serves as a useful resource for researchers and practitioners, guiding further investigation and application of AI and deep learning in this domain.

Furthermore, our review acknowledges a notable gap in addressing perspectives from patients and clinicians regarding the use of AI in spinal oncology. Incorporating feedback from these key stakeholders is essential for understanding the real-world implications of AI applications. Patients’ experiences [220] and clinicians’ insights [221,222] are critical for evaluating the practical benefits and limitations of AI technologies, as well as their acceptance and integration into routine clinical practice. Additionally, our review did not extensively cover the long-term impact and cost-effectiveness of AI applications [223]. Evaluating these factors is important for assessing the overall value of AI to spinal oncology, as it can influence decision-making processes, resource allocation, and the sustainability of these technologies in clinical settings. Future research should address these aspects to provide a more comprehensive evaluation of AI’s role in spinal oncology, ensuring that the benefits of AI applications are balanced with considerations of their economic and long-term impacts.

## 6. Conclusions

In conclusion, the integration of deep learning techniques with computed tomography (CT) imaging in spinal oncology presents a promising avenue for enhancing diagnostic accuracy, treatment planning, and improving patient outcomes. This review has demonstrated the diverse applications of artificial intelligence (AI) in CT imaging of spinal metastases, including detection, classification, grading, and treatment planning. AI technologies have demonstrated notable performance across these domains, offering the potential to support clinicians by improving workflow efficiency and minimizing complications. However, despite promising findings, additional research is warranted to validate the clinical effectiveness of these AI tools and streamline their integration into everyday clinical workflows. Ultimately, the continued exploration and refinement of AI applications in CT spinal imaging holds immense promise for advancing the field of spinal oncology and improving patient care.

## Figures and Tables

**Figure 1 cancers-16-02988-f001:**
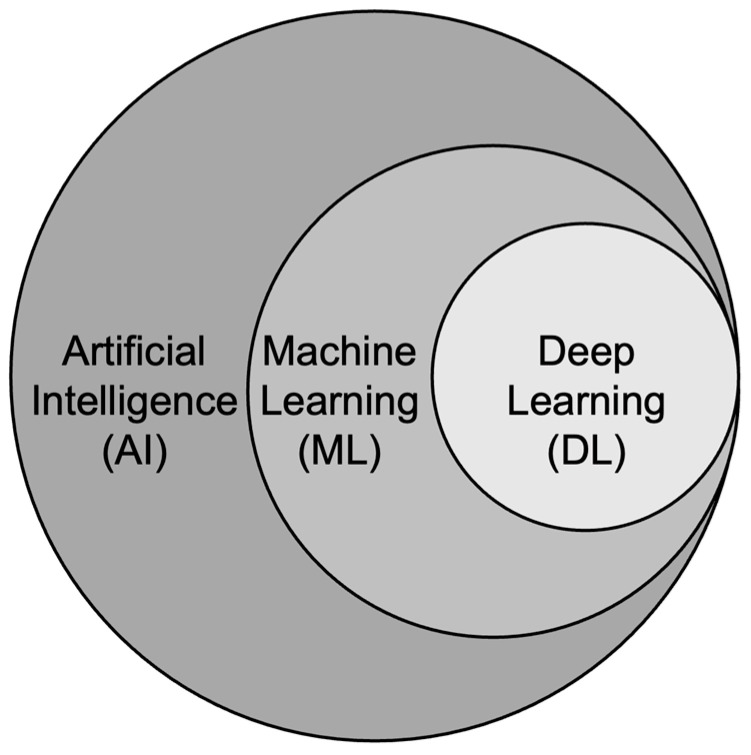
Figure depicting the hierarchical arrangement of artificial intelligence (AI). Machine learning (ML), a subset of AI, aims to empower computers to learn independently without explicit programming. Deep learning (DL), a specialized field within machine learning, involves the computation of neural networks comprising multiple layers.

**Figure 2 cancers-16-02988-f002:**
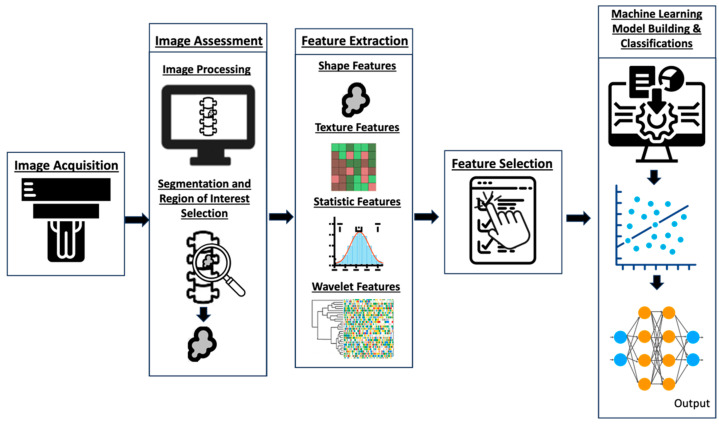
Diagram illustrating the basic framework and essential stages of radiomics, including image acquisition, image processing (including segmentation), feature extraction within specified regions of interest (ROIs), feature selection, exploratory analysis, and subsequent modeling.

**Figure 3 cancers-16-02988-f003:**
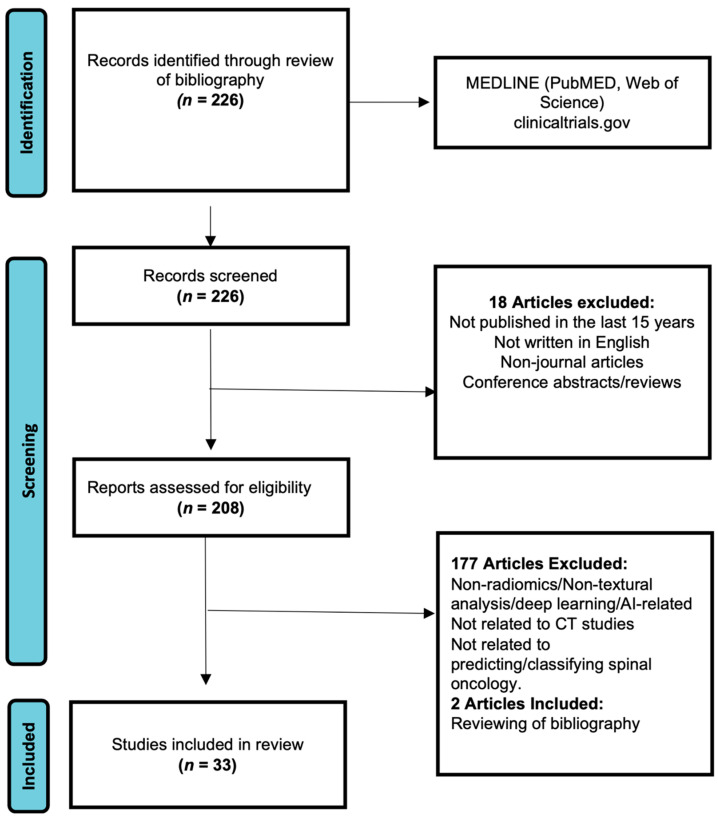
PRISMA flowchart (adapted from PRISMA group, 2020) outlining the process of selecting pertinent articles for analysis.

**Figure 4 cancers-16-02988-f004:**
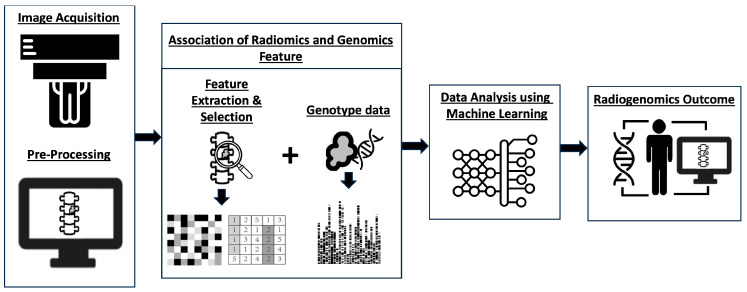
Diagram illustrating a typical radiogenomics process, encompassing image acquisition and pre-processing, feature extraction and selection from medical imaging and genotype data, correlation between radiomic and genomic features, data analysis using machine learning models, and the determination of the final radiogenomics outcome.

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
