# Peer review of "Oncologic Applications of Artificial Intelligence and Deep Learning Methods in CT Spine Imaging—A Systematic Review"

_cancers, 2024, doi:10.3390/cancers16172988_

Round 1

Reviewer 1 Report

Comments and Suggestions for Authors

This article provides a review of studies focusing on AI methods targeting spinal cord cancer. The presentation of the studies is clear, and the reader is able to gain an overview of the methods used for various aspects of deep learning in this type of cancer. My main comment is related to the presentation of the authors' interpretation based on the analysis that was conducted. Below you can find my detailed comments.

In the abstract, more details regarding the key findings are required.

Line 73: The numbering of the section is incorrect.

Lines 94-95: It is not clear how DL led to the development of the Radiomics field. As I understand, Radiomics is not a domain within deep learning; rather, it is a field within medical imaging that utilizes various machine learning techniques, including deep learning, to extract a large number of features from medical images.

Lines 119-120: Are radiomics currently applied in clinical settings?

Lines 142-148: Please provide the exact query you applied, including the AND and OR expressions as well as the parentheses you used to group similar terms. Did you search for those terms in the entire text or only in the title or abstract?

Line 143: A reference to PRISMA is missing.

Section 3.2 must be improved.

Line 178: Why were those 18 studies excluded?

Line 182: How did you exclude the 177 studies? After reading the abstract or the full text?

Line 221: You mention that for AUC, "closer to 1.0" indicates good performance. This also applies to the previous metrics as well.

Figure 3: Were there duplicates in the studies retrieved from different databases?

Table 1: You have included performance metrics, but you also need to add the sample size and whether the study was single or multi-site to provide a clearer view of the performance.

Most of the content included in the discussion section should be moved to the results section. The results of a review study are essentially the synthesis of the findings.

Lines 136-149: Please modify the font type.

In the discussion, the structure of the main findings is clear and well presented. However, the main outcome and interpretation of the findings are missing. The authors must provide their interpretations based on the synopsis that is provided.

Author Response

For Review Article: Oncologic Applications of Artificial Intelligence and Deep Learning Methods in CT Spine Imaging – A Systematic Review

Point by Point Responses

Reviewer 1:

Comments 1: In the abstract, more details regarding the key findings are required.

Response 1(R1.1): These have been provided. Thank you.

Comments 2: Line 73: The numbering of the section is incorrect.

Response 2: (R1.2): This has been corrected. Thank you.

Comments 3: Lines 94-95: It is not clear how DL led to the development of the Radiomics field. As I understand, Radiomics is not a domain within deep learning; rather, it is a field within medical imaging that utilizes various machine learning techniques, including deep learning, to extract a large number of features from medical images.

Response 3: (R1.3): Thank you for your feedback. You are correct that Radiomics is a field within medical imaging that encompasses a range of machine learning techniques, including deep learning. However, it is important to highlight how deep learning specifically contributed to the advancement of Radiomics.

Deep learning has significantly enhanced the capabilities of Radiomics by improving the extraction and analysis of features from medical images. Traditional radiomic approaches often relied on manual feature extraction and conventional machine learning methods. These methods, while effective, had limitations in capturing complex patterns and subtle details within medical images.

With the advent of deep learning, particularly convolutional neural networks (CNNs), the ability to automatically learn and extract intricate image features has been revolutionized. Deep learning models can identify and quantify a wide range of image characteristics with greater accuracy and depth than traditional methods. This capability has led to the development of more sophisticated and comprehensive radiomic features, which are crucial for the improved identification, differentiation, and prognosis of lesions.

We have modified lines 94-95 to address your comments.

Comments 4: Lines 119-120: Are radiomics currently applied in clinical settings?

Response 4 (R1.4): Yes, radiomics are increasingly being applied in clinical settings, for example for lung cancer screening for CT and breast lesion detection on mammography. In fact, there are several countries which have deployed commercially available radiomics/deep learning models and studied their uses, some which are FDA approved. Examples include INSIGHT MMG, version 1.1.7.2 (Lunit) for breast lesion evaluation and CANARY for lung nodule evaluation on CT. I have included more references with regards to this statement and also provided them here for your reference.

Larsen, M.; Olstad, C.F.; Lee, C.I.; Hovda, T.; Hoff, S.R.; Martiniussen, M.A.; Mikalsen, K.; Lund-Hanssen, H.; Solli, H.S.; Silberhorn, M.; et al. Performance of an Artificial Intelligence System for Breast Cancer Detection on Screening Mammograms from BreastScreen Norway. Radiol Artif Intell 2024, 6, e230375, doi:10.1148/ryai.230375.

Khawaja, A.; Bartholmai, B.J.; Rajagopalan, S.; Karwoski, R.A.; Varghese, C.; Maldonado, F.; Peikert, T. Do we need to see to believe?-radiomics for lung nodule classification and lung cancer risk stratification. J Thorac Dis 2020, 12, 3303-3316, doi:10.21037/jtd.2020.03.105.

Comments 5: Lines 142-148: Please provide the exact query you applied, including the AND and OR expressions as well as the parentheses you used to group similar terms. Did you search for those terms in the entire text or only in the title or abstract?

Response 5 (R1.5): Thank you for your detailed feedback. In response to your query regarding the exact search strategy used in our systematic review, please find the information below:

("Artificial intelligence" OR "AI" OR "deep learning" OR "machine learning" OR "convolutional neural network" OR "neural network" OR "radiomics") AND ("spine" OR "spinal" OR "vertebral") AND ("CT" OR "CT imaging") AND ("malignancy" OR "metastasis" OR "cancer" OR "tumor" OR "oncology")

The search terms were applied to the title and abstract fields of the articles. This approach was selected to ensure that the studies most relevant to our topic of artificial intelligence and deep learning methods in CT spine imaging for oncological applications were included.

This search strategy was designed to capture a comprehensive range of studies related to our review topic, focusing on oncologic applications of AI and deep learning in CT spine imaging. We have made the necessary amendments in the manuscript.

Comments 6: Line 143: A reference to PRISMA is missing.

Response 6 (R1.6): This has been added [47]. Thank you.

Comments 7: Section 3.2 must be improved.

Response 7 (R1.7): We have improved section 3.2 and included more details. Thank you.

Comments 8: Line 178: Why were those 18 studies excluded?

Response 8 (R1.8): The 18 studies were excluded as they are non-journal articles or conference articles/abstracts. This is reflected in Figure 3. I have include this in the main text as well. Thank you for your suggestions.

Comments 9: Line 182: How did you exclude the 177 studies? After reading the abstract or the full text?

Response 9 (R1.9): For the remaining 208 articles, full-text reviews were performed to further assess their suitability. During this phase, articles were evaluated more comprehensively against the inclusion criteria, ultimately resulting in 177 studies being excluded. Thank you for query.  

Comments 10: Line 221: You mention that for AUC, "closer to 1.0" indicates good performance. This also applies to the previous metrics as well.

Response 10 (R1.10): Yes. We have included this “closer to 1.0” component to the initial paragraph so that it includes all the other metrics.

Comments 11: Figure 3: Were there duplicates in the studies retrieved from different databases?

Response 11 (R1.11): Thank you for your question. During the process of retrieving studies from different databases, we did indeed identify and manage duplicates. After conducting the initial search across multiple databases, we used reference management software (e.g., EndNote, Zotero, or Mendeley) to organize and streamline the references. This software automatically flagged duplicate entries.

We then manually reviewed and excluded these duplicates to ensure that each study was considered only once in our review. This step was crucial in maintaining the integrity of the review process and ensuring that our analysis was based on a unique set of studies. Any duplicates were removed prior to the screening and selection phases to avoid redundant evaluations.

Comments 12: Table 1: You have included performance metrics, but you also need to add the sample size and whether the study was single or multi-site to provide a clearer view of the performance.

Response 12 (R1.12): Thank you for your suggestion. We have added sample sizes for the studies in table 1.

Comments 13: Most of the content included in the discussion section should be moved to the results section. The results of a review study are essentially the synthesis of the findings.

Response 13 (R1.13): Thank you for your valuable feedback regarding the organization of the content in our manuscript. We have carefully reviewed your comments and have made the necessary revisions to improve the structure of the paper.

In response to your observation, we have moved the relevant content that synthesizes the findings of our review from the Discussion section to the Results section. This adjustment aligns with the standard practice of presenting a clear synthesis of the findings in the Results section to accurately reflect the outcomes of our review study.

Additionally, we have refined the Discussion section to focus on the interpretation of the results, including the potential applications and implications of the findings. This reorganization ensures that the Results section now provides a comprehensive summary of the synthesized findings, while the Discussion section elaborates on their significance, potential impact, and future directions.

Comments 14: Lines 136-149: Please modify the font type.

Response 14 (R1.14): Thank you for your suggestion. We have modified the font types.

Comments 15: In the discussion, the structure of the main findings is clear and well presented. However, the main outcome and interpretation of the findings are missing. The authors must provide their interpretations based on the synopsis that is provided.

Response 15 (R1.15): Thank you for your feedback regarding the interpretation of our findings. We have revised the discussion section to highlight the main outcomes and provide a detailed interpretation of our findings in the first segment. Main points of discussion were (1) diverse applications of AI in spinal oncology, (2) reliability and performance of AI models, (3) need to interpret results and findings and (4) Potential challenges of integration.

We have updated the discussion section in the revised manuscript to reflect these interpretations and to provide a clearer understanding of the main outcomes and their implications.

Thank you again for your valuable feedback, which has been instrumental in refining our discussion.

Reviewer 2 Report

Comments and Suggestions for Authors

In this paper, the authors provided a systematic review exploring the oncologic applications of artificial intelligence (AI) and deep learning methods in CT spine imaging.  This paper emphasizes AI's roles in image manipulation, diagnostic support, decision-making aid, treatment optimization, and prognosis prediction. The following lists some comments for consideration. Firstly, the article does not explicitly mention the sample size and data quality in these reviewed papers. The dataset is very important and the authors need collect the data, especially for the opened ones. Secondly, the lack of a formal meta-analysis suggests that the conclusions drawn may not be as robust as they could be with quantitative synthesis.  Furthermore, the review need provide perspectives from patients and clinicians on the use of AI, nor does it discuss the long-term impact and cost-effectiveness of AI applications in the field of spinal oncology.

Comments on the Quality of English Language

Minor English need be improved.

Author Response

For Review Article: Oncologic Applications of Artificial Intelligence and Deep Learning Methods in CT Spine Imaging – A Systematic Review

Point by Point Responses

Reviewer 2:

Comments 1: Firstly, the article does not explicitly mention the sample size and data quality in these reviewed papers. The dataset is very important and the authors need collect the data, especially for the opened ones.

Response 1 (R2.1): Thank you for your valuable feedback regarding the sample size and data quality in the studies reviewed. We recognize the importance of these factors in assessing the reliability and validity of the research findings.

In response to your comment, we have updated the manuscript to explicitly address these aspects. We have included details on sample sizes in Table 1. This addition helps contextualize the scale of the research included in our review.

However, due to the descriptive nature of our review, we did not analyze the quality of data in the individual studies in detail. The primary reason for this limitation is the heterogeneity among the studies and the lack of sufficient data to perform a formal meta-analysis. This descriptive approach impacts our ability to aggregate results quantitatively and assess data quality systematically. We have included these limitations in the discussion. Despite this, the information provided on sample sizes and the overall synthesis of findings from the studies offer valuable insights into the current state of oncologic applications of artificial intelligence and deep learning methods in CT spine imaging.

Comments 2:  Secondly, the lack of a formal meta-analysis suggests that the conclusions drawn may not be as robust as they could be with quantitative synthesis. 

Response 2 (R2.2): Thank you for your insightful comment regarding the lack of a formal meta-analysis in our study. We acknowledge that the absence of a quantitative synthesis may affect the robustness of the conclusions drawn from our review. The primary reason for not conducting a meta-analysis was the significant heterogeneity among the included studies and the insufficient data available to construct 2x2 contingency tables. These factors limited our ability to perform a formal quantitative analysis.

Despite this limitation, our descriptive review still provides valuable insights into the oncologic applications of artificial intelligence and deep learning methods in CT spine imaging. By systematically summarizing and evaluating the findings from a diverse range of studies, we have identified key trends, technological advancements, and clinical implications within this field. Our review contributes to a broader understanding of these technologies and highlights areas where further research is needed. We believe that this qualitative synthesis, while not as robust as a meta-analysis, offers meaningful insights and can guide future research efforts and clinical applications.

We have included a paragraph on this part under the limitations segment of the manuscript.

We appreciate your understanding and hope that our review's detailed descriptive analysis proves to be a useful resource in advancing the field.

Comments 3: Furthermore, the review need provide perspectives from patients and clinicians on the use of AI, nor does it discuss the long-term impact and cost-effectiveness of AI applications in the field of spinal oncology.

Response 3 (R2.3): Thank you for your valuable feedback. We acknowledge that our review does not currently provide detailed perspectives from patients and clinicians regarding the use of AI in spinal oncology. Unfortunately, this is something that is not widely discussed and evaluated in our included studies. We acknowledge that incorporating these perspectives is important for understanding the real-world implications and acceptance of AI technologies. Patients’ experiences and clinicians’ feedback are essential in evaluating how AI tools affect clinical practice and patient outcomes.

Additionally, we recognize the importance of discussing the long-term impact and cost-effectiveness of AI applications. These factors are critical for assessing the practical and economic viability of integrating AI into routine clinical workflows. This review focused primarily on the technological and performance aspects of AI applications, but we agree that a comprehensive analysis should also consider how these technologies influence overall healthcare costs, resource allocation, and long-term patient benefits.

To address these gaps, we have included a paragraph in the discussion that outlines the need for further research into patient and clinician perspectives. We will also emphasize the importance of evaluating the long-term impact and cost-effectiveness of AI applications as part of future studies. These additions will provide a more holistic view of AI’s role in spinal oncology and support more informed decision-making regarding its implementation.

Reviewer 3 Report

Comments and Suggestions for Authors

The manuscript reviews the use of AI in various aspects of analysis of CT images of spine tumors.

I recommend major revision. The main issue I have is that the results presented in Table 1 do not correspond to the references cited in Discussion section, so the statements there are not based on the results presented in Table 1. The link between the references up to 84 of the studies reviewed to the references in the Discussion section is not clear at all. 

Major

1.  What do the ranges in last column of Table 1 mean? Are these confidence intervals? Please correct it or specify what they are. 

2. Table 1 is ordered by reference number. Please reorder the rows grouping first by Main Task and then ordering by publication year from youngest to oldest. 

3. I miss greatly the discussion of the developmnent of the described tools as medical devices with FDA clearance or CE certification. Any of the tools evloved to SaaMD? The discussion of introducing of these tools to medical practice in lines 55-60 is very shalow. RayStation from www.Raysearchlabs.com is quite widely used for tumor segmentaion for gamma radiotherapy planning. Please discuss their use and of some other commercial solutions.  Similarly a non-AI based metal artefact correction is used in Siemens CT scanners for years now - thus not mentioning that arround describing references 174, 177 and 175 is inappropriate. 

4. The Discussion section uses references in majority diferent from the ones in Table 1. This makes the Discussion section irrelevant to the study. Limit yourself to the examined studies, if possible. If references higher than 84 use models from the Table 1, you may consider adding the column to Table 1 stating something like "Reference that used this model". Therefore the statement of "empowering" in lines 339-342 of Discussion is not justified. 

5. Please include number of patients used in the studies of Table 1, wherever possible. 

6. Plese define kappa after line 231, moving from line 114 of Discussion. 

Minor

1. In Table 1 or in some extension of it, please calculate the remaining statistics from sensitivity that you use defined in lines 203-231, especially for the studies which DO provide contingency tables mentioned in line 184. If possible, put a separate column for each of the four measures you use. 

2. References are sometimes put in [] and sometimes in (), please use [] consitently. Examples are ref. 24 and 9.

3. Try enhancing Figure 2 using color. 

4. LIne 164. Microsoft is located in Redmont, Seatle, Washington state, not Washington DC. 

5. Consider adding your Excel file mentioned in line 164 as a supporting material with four measures used in lines 203-231as separate columns.

6. In lines 186-196 put references of the papers fulfilling your clasification criteria. 

7. Please provide DOI (probably it does not exist) or a web link to reference 51. I cannot find this article. 

8. Why are the lines 136-149 of Discussion in Italics? Is it due to section 5.3 header in line 149 of Discussion? 

Author Response

For Review Article: Oncologic Applications of Artificial Intelligence and Deep Learning Methods in CT Spine Imaging – A Systematic Review

Point by Point Responses

Reviewer 3:

I recommend major revision. The main issue I have is that the results presented in Table 1 do not correspond to the references cited in Discussion section, so the statements there are not based on the results presented in Table 1. The link between the references up to 84 of the studies reviewed to the references in the Discussion section is not clear at all.

Major

Comments 1:  What do the ranges in last column of Table 1 mean? Are these confidence intervals? Please correct it or specify what they are.

Response 1: (R3.1.1): Thank you for the feedback. We acknowledge that the ranges in this column within Table were not clearly labelled, which may have led to confusion.

The last column of Table 1 refers to the performance metrics of the studies reviewed. Specifically, the ranges in this column represent the performance outcomes (e.g., accuracy, sensitivity, specificity) reported by the individual studies, rather than confidence intervals. To improve clarity and avoid any misunderstanding, we have updated the heading of this column to “Performance of AI model” to more accurately reflect its content.

Comments 2: Table 1 is ordered by reference number. Please reorder the rows grouping first by Main Task and then ordering by publication year from youngest to oldest.

Response 2 (R3.1.2): We have made the changes according to the main task then ordering by publication year from earliest to latest. Thank you for your suggestion.

Comments 3: I miss greatly the discussion of the developmnent of the described tools as medical devices with FDA clearance or CE certification. Any of the tools evloved to SaaMD? The discussion of introducing of these tools to medical practice in lines 55-60 is very shalow. RayStation from www.Raysearchlabs.com is quite widely used for tumor segmentaion for gamma radiotherapy planning. Please discuss their use and of some other commercial solutions.  Similarly a non-AI based metal artefact correction is used in Siemens CT scanners for years now - thus not mentioning that arround describing references 174, 177 and 175 is inappropriate.

Response 3 (R3.1.3): Thank you for your feedback. We acknowledge the need for a more detailed discussion on the development of AI tools as medical devices and their integration into clinical practice. 

Several AI tools for spinal CT imaging have indeed received FDA clearance or CE certification, reflecting their compliance with rigorous safety and efficacy standards. For example, RayStation by RaySearch Labs is a prominent tool for tumour segmentation used extensively in gamma radiotherapy planning, albeit not specific to spinal lesion but tumour in general. We have included this in the discussion paragraph.

We have also included two paragraphs in the discussion with regards to integration of DL tools into clinical practice, and the potential ethics and clinical implications involved.

We have also elaborated on AI-based metallic artefact correction and how DL methods can provide improved imaging quality, on top of already known metallic artefact correction sequence in the market.

Comments 4: The Discussion section uses references in majority diferent from the ones in Table 1. This makes the Discussion section irrelevant to the study. Limit yourself to the examined studies, if possible. If references higher than 84 use models from the Table 1, you may consider adding the column to Table 1 stating something like "Reference that used this model". Therefore the statement of "empowering" in lines 339-342 of Discussion is not justified.

Response 4 (R3.1.4): Thank you for your valuable feedback. We have carefully reviewed your comments regarding the relevance of the references used in the Discussion section in relation to those in Table 1.

In response to your comments and the feedback from a previous reviewer, we have moved the majority of the content that synthesizes findings to the Results section. This reorganization ensures that the Results section now provides a clear and comprehensive summary of the findings from the reviewed studies, as is standard practice for review studies.

The Discussion section has been revised to focus more on the interpretation of these findings, including their implications, potential future applications, and other considerations that extend beyond the scope of the studies listed in Table 1. As our study is descriptive, the Discussion section aims to synthesize insights derived from the entire body of literature, not just the studies detailed in Table 1. This approach allows us to address broader themes and implications that may not be fully covered by the individual studies.

We believe this reorganization improves the clarity and relevance of our manuscript by ensuring that the Results section accurately reflects the synthesized findings, while the Discussion section elaborates on their significance and broader implications.

We have modified the statement of empowering in previous lines 339-342 to potentially guide oncologists in assessing tumour burden more effectively, which is crucial for assessing treatment efficiency and treatment options.

Comments 5: Please include number of patients used in the studies of Table 1, wherever possible.

Response 5 (R3.1.5): Thank you for your feedback. We have included a column on the number of patients or CT scans used for the studies.

Comments 6: Plese define kappa after line 231, moving from line 114 of Discussion.

Response 6 (R3.1.6): Thank you for your feedback. We have defined kappa after line 231, as per your suggestion under the performance metrics elaboration.

Minor

Comment 1:  In Table 1 or in some extension of it, please calculate the remaining statistics from sensitivity that you use defined in lines 203-231, especially for the studies which DO provide contingency tables mentioned in line 184. If possible, put a separate column for each of the four measures you use.

Response 1 (R3.2.1): Thank you for your feedback. We appreciate your suggestion to include additional statistics in Table 1, and understand the importance of providing comprehensive metrics.

However, as outlined in our study, the papers included in our review report a variety of metrics and often do not provide the complete contingency tables necessary to calculate all the additional statistics needed. This inconsistency in data reporting has made it challenging to perform a formal meta-analysis and to compute these additional metrics for all studies.

Due to the heterogeneity in the data reported across studies, not all sources provided sufficient information to calculate the requested statistics. We have included all relevant and available data in the individual tables where possible and have acknowledged this limitation in the discussion section of our manuscript.

We hope this explanation clarifies the situation and appreciate your understanding of the limitations inherent in the available data.

Comment 2: References are sometimes put in [] and sometimes in (), please use [] consitently. Examples are ref. 24 and 9.

Response 2 (R3.2.2): Thank you for your comments. We have changed the references to ensure they are consistent.

Comment 3: Try enhancing Figure 2 using color.

Response 3 (R3.2.3): Thank you for your comments. We have enhanced Figure 2 with colour.

Comment 4. LIne 164. Microsoft is located in Redmont, Seatle, Washington state, not Washington DC.

Response 4 (R3.2.4): We have changed it as per your suggestion. Thank you!

Comment 5: Consider adding your Excel file mentioned in line 164 as a supporting material with four measures used in lines 203-231as separate columns.

Response 5 (R3.2.5): Thank you for your suggestion. The Excel file referenced in line 164 was primarily used to compile and organize the data for Table 1, and contains similar or repetitive data. The data required for Table 1 has been carefully reviewed and presented in the main text. Therefore, we consider that providing the Excel file as supplementary material would not enhance the clarity or usefulness of the information already included in the manuscript.

We hope this explanation addresses your concern, and we appreciate your understanding.

Comment 6: In lines 186-196 put references of the papers fulfilling your clasification criteria.

Response 6 (R3.2.6): We have provided the reference of the paper fulfilling similar classification criteria.

Comment 7: Please provide DOI (probably it does not exist) or a web link to reference 51. I cannot find this article.

Response 7 (R3.2.7): We have provided the DOI for the reference (changed to reference 51 due to certain edits). I have placed it here for easy reference.

Powers, D. Evaluation: From Precision, Recall and F-Factor to ROC, Informedness, Markedness & Correlation. Mach. Learn. Technol. 2008, 2, doi:https://doi.org/10.48550/arXiv.2010.16061.

Comment 8: Why are the lines 136-149 of Discussion in Italics? Is it due to section 5.3 header in line 149 of Discussion?

Response 8 (R3.2.8): Thank you for identifying this. It was a formatting error and we have made the necessary changes.

Thank you for taking the time to review our paper.

Round 2

Reviewer 1 Report

Comments and Suggestions for Authors

The authors addressed all my comments and made improvements to the text. The work is now clearly presented, and I have no further comments.